# Characterizing epidemiology of prediabetes, diabetes, and hypertension in Qataris: A cross-sectional study

**Mohammed H. Al-Thani[1], Kholood A. Al-Mutawa[1], Salah A. Alyafei[1], Muhammad A. Ijaz[1], Shamseldin A. H. Khalifa[1], Suresh B. Kokku[1], Amit C. M. Mishra[1], Benjamin V. Poovelil[1], Mounir B. Soussi[1], Amine A. Toumi[1], Soha R. Dargham[2,3], Susanne F. Awad[2,3,4], Laith J. Abu-Raddad[2,3,4,5] \***

**1** Public Health Department, Ministry of Public Health, Doha, Qatar, **2** Infectious Disease Epidemiology Group, Weill Cornell Medicine—Qatar, Cornell University, Doha, Qatar, **3** World Health Organization Collaborating Centre for Disease Epidemiology Analytics on HIV/AIDS, Sexually Transmitted Infections, and Viral Hepatitis, Weill Cornell Medicine–Qatar, Cornell University, Doha, Qatar, **4** Department of Population Health Sciences, Weill Cornell Medicine, Cornell University, New York, New York, United States of America, **5** Department of Public Health, College of Health Sciences, QU Health, Qatar University, Doha, Qatar

\* lja2002@qatar-med.cornell.edu

**Data Availability Statement:** The dataset of this study is a property of the Qatar Ministry of Public Health that was provided to the researchers through a restricted-access agreement that

## Abstract

### Objectives

To characterize the epidemiologic profiles of prediabetes mellitus (preDM), diabetes mellitus (DM), and hypertension (HTN) in Qataris using the nationally representative 2012 Qatar STEPwise Survey.

### Methods

A secondary data analysis of a cross-sectional survey that included 2,497 Qatari nationals aged 18–64 years. Descriptive and analytical statistical analyses were conducted.

### Results

Prevalence of preDM, DM, and HTN in Qataris aged 18–64 years was 11.9% (95% confidence interval [CI] 9.6%-14.7%), 10.4% (95% CI 8.4%-12.9%), and 32.9% (95% CI 30.4%-35.6%), respectively. Age was the common factor associated with the three conditions. Adjusted analyses showed that unhealthy diet (adjusted odds ratio (aOR) = 1.84, 95% CI 1.01–3.36) was significantly associated with preDM; that physical inactivity (aOR = 1.66, 95% CI 1.12–2.46), central obesity (aOR = 2.08, 95% CI 1.02–4.26), and HTN (aOR = 2.18, 95% CI 1.40–3.38) were significantly associated with DM; and that DM (aOR = 2.07, 95% CI 1.34–3.22) was significantly associated with HTN. Population attributable fraction of preDM associated with unhealthy diet was 7.7%; of DM associated with physical inactivity, central obesity, and HTN, respectively, was 14.9%, 39.8%, and 17.5%; and of HTN associated with DM was 3.0%.

prevents sharing the dataset with a third party or publicly. Future access to this dataset can be considered through a direct application for data access to Her Excellency Dr. Hanan Mohammed Al Kuwari, the Minister of Public Health (https://www. moph.gov.qa/english/Pages/default.aspx; e-mail: GHCC@moph.gov.qa; phone: +974 44070000). Aggregate data are available within the manuscript and its Supplementary information.

**Funding:** This publication was made possible by NPRP grant number 10-1208-160017 from the Qatar National Research Fund (a member of Qatar Foundation). The statements made herein are solely the responsibility of the authors. Please note that the authors amended the FD statement: The funder had no role in study design, data collection and analysis, decision to publish, or preparation of the manuscript.

**Competing interests:** The authors have declared that no competing interests exist.

## Conclusions

One in five Qataris is living with either preDM or DM, and one in three is living with HTN, conditions that were found to be primarily driven by lifestyle factors. Prevention, control, and management of these conditions should be a national priority to reduce their disease burden and associated disease sequelae.

## Introduction

Abnormalities in the metabolic syndrome, such as prediabetes mellitus (preDM), diabetes mellitus (DM), and hypertension (HTN), are key causes of cardiovascular diseases (CVDs) and of disproportionately large number of premature deaths globally [1–3]. Mechanisms, such as endothelial dysfunction, oxidative stress, and vascular inflammation are primary causes of CVDs in people with DM and HTN [4,5]. An estimated 9.2%, 12.8%, and 29.5% of the population of the Middle East and North Africa (MENA) are living with preDM, DM, and HTN, respectively [1,6]. Although these estimates are high, the epidemiological profiles of these conditions in MENA remain insufficiently characterized [6]. Nonetheless, it is evident from published global literature that anthropometric and behavioral risk factors, such as physical inactivity, smoking, diet, obesity, and family history of DM and HTN are important risk factors for developing non-communicable diseases (NCDs) including preDM, DM, and HTN [1,7–14].

Identifying the need for reliable data on key risk factors for NCDs, the World Health Organization (WHO) initiated a STEPwise approach to surveillance (STEPS) to facilitate the elucidation of population profiles of NCD risk-factors, and to evaluate and inform policies and programs across countries [15,16]. Qatar, a MENA country, used the STEPS approach to describe the burden of preDM, DM, and HTN. In the 2012 Qatar STEPwise Survey—the first and still the only nationally representative population-based survey conducted for these outcomes—preDM, DM, and HTN prevalences among adult Qataris aged 18–64 years were reported at 5.8% (95% confidence interval [CI] 4.5%-7.2%), 16.7% (95% CI 13.7%–19.8%), and 32.9% (95% CI 30.2%-35.6%), respectively [17].

To estimate the burden of preDM and DM, the 2012 Qatar STEPwise Survey employed the cut-off values of 100 mg/dL to <110 mg/dL for preDM and ≥110 mg/dL for DM [17]. These cut-offs were applied to blood glucose levels measured using a point-of-care (POC) device by sampling blood from capillary whole-blood rather than employing the standard laboratory testing of centrifuged plasma sampled from venous blood [17]. A recent global analysis of STEPS surveys identified this application of POC-device cut-offs to be inaccurate, and since 2010, devices used for capillary whole blood tests have been internally calibrated to produce plasma-equivalent glucose measurements [18]. Accordingly, the POC glucose measuring device and test strips used in the 2012 Qatar STEPwise Survey were also plasma calibrated (S1 Appendix). Incorrect fasting glucose thresholds were employed to define preDM and DM instead of the appropriate thresholds of 110 mg/dL to <126 mg/dL and ≥126 mg/dL [18]. Consequently, the prevalence reported for Qataris could be artefactually deflated or inflated for both preDM and DM.

Against this background, we aimed to update and characterize the epidemiologic profiles of preDM, DM, and HTN among Qataris aged 18–64 years using the data from the nationally representative 2012 Qatar STEPwise Survey [17]. Specifically, for preDM and DM, we updated the prevalence estimates using the appropriate preDM and DM cut-offs [18]. We also

investigated factors possibly associated with these outcomes. While the prevalence estimate for HTN has been reported earlier in the 2012 Qatar STEPwise Survey report [17], in this study we complemented this report with regressions to investigate factors associated with HTN. Lastly, we also estimated the population attributable fraction for the statistically significant and modifiable factors associated with preDM, DM, and HTN. The study was conducted to inform public health policy, programming, and resource allocation.

## Material and methods

### Data source

This is a cross-sectional study using data from the 2012 Qatar STEPwise Survey [17], conducted in accordance with the WHO's STEPS approach [15]. This approach was based on standardized questions and protocols to collect data on prevalence and known key risk factors of NCDs in a population [15]. The STEPS instrument covered three different modules of risk factor assessment: self-reported questionnaire (Step 1), physical measurements (Step 2), and biochemical measurements (Step 3).

Step 1 presented information on demographic characteristics, tobacco use, physical activity, and medical history (self-reported history of DM, HTN, and medication use). Step 2 reported measurements of height, weight, waist and hip circumference, and blood pressure. Step 3 examined blood glucose and lipid levels in the collected capillary whole blood sample. The data extraction tool was a questionnaire written in Arabic with an English version as well and can be found in the 2012 Qatar STEPwise Survey report [17] (English version is attached in S2 Appendix).

### Sampling and participants

The detailed description of the 2012 Qatar STEPwise Survey design and recruitment is in Qatar's 2012 STEPwise report [17]. Briefly, the survey was conducted between March and May 2012 on only Qatari households who represent the permanent population of Qatar. The sampling was done in a two-stage cluster strategy. In the first one, 96 primary selected units (PSUs) were selected based on probability-based sampling that is proportional to the 2010 population size. In the second stage, a sample of 30 households was selected within each PSU using systematic random sampling.

A total of 2,497 Qataris aged 18–64 years from seven municipalities—with each municipality representing an administrative structure within Qatar—participated in the survey. The participation rate was 88%. All the participants were adequately informed about the survey and provided written consent prior to study enrolment. The study was approved by the ethics board at the Ministry of Public Health. All data used in the present study were de-identified and anonymized.

### Outcome measures and associated factors

Diagnostic criteria of each outcome (preDM, DM, and HTN) and associated factors (physical inactivity, diet, obesity, and central obesity) were based on WHO definitions [17–24]. Two blood pressure measurements were taken; if they differed by more than 25/15 mmHg then a third measurement was taken. The average of the blood pressure measurements was used in the analysis. HTN was defined as systolic blood pressure ≥140 mmHg and/or a diastolic blood pressure ≥90 mmHg [20], or use of HTN medication prescribed by a doctor in the previous two weeks. As indicated above, preDM was defined as fasting glucose ≥110 mg/dL to <126 mg/dL [18,19], while DM was defined as fasting glucose ≥126 mg/dL [18,19], or use of insulin

or DM medication at the day of the fasting blood glucose test. The POC device, CardioChek PA analyzer, and PTS Panels test strips manufactured in 2012 were utilized for measuring blood glucose levels. The device could detect glucose levels in the range of 20 mg/dL to 600 mg/dL. For levels outside this range, the test was to be repeated using the same device. The Omron BP785 digital device was used to measure blood pressure levels [17]. These devices were recommended by the WHO [17]. Detailed description of the 2012 version of the POC device was provided by the manufacturer and is included in S1 Appendix. The attached 2012 version of the insert was valid at the time of the conduct of the survey but has since been superseded.

Three education levels were included in the analyses: completion of university degree, completion of preparatory or secondary school, and completion of primary school or less. Marital status was dichotomized into currently married or non-married (i.e., single, divorced, or widowed). Current smoking status was defined as yes (current use of any tobacco products) and no (past- or non-smoker). Physical inactivity was defined as <150 minutes of moderate activity and <75 minutes of vigorous activity per week (i.e. <600 metabolic equivalent-minutes per week) [17,24,25]. Diet was classified as unhealthy (consumption of <5 portions of fruits and vegetables per day) and healthy (≥5 portions per day) [21]. Obesity was defined as body mass index ≥30 [22], while central obesity was defined as a waist circumference ≥94 cm for men and ≥80 cm for women [23]. Family history of DM was defined as at least one of the parents, children, brothers, or sisters is living with DM, while family history of HTN was defined as at least one of the parents, brothers, or sisters, is living with HTN [17].

## Statistical methods

The characteristics of the sample were summarized using frequency distributions. Age was stratified into 5 age groups: ≤24, 25–34, 35–44, 45–54, and 55–64. Age- and sex-specific distributions of preDM, DM, and HTN were presented. P-trend, denoting the p-value for a trend, was reported when investigating a linear trend using the Cochran-Armitage (linear-by-linear association) test. To avoid low power when performing the logistic regressions, we recategorized age into three groups: <30, 30–49, and ≥50 years. For each outcome, univariate and multivariable logistic regressions were conducted to investigate factors associated with the outcomes. For the multivariable logistic regression, all factors associated with a crude p-value <0.200 were accounted for. A sensitivity analysis was also conducted adding sex to the multivariable logistic regression for relevance despite its crude p-value>0.200. Unadjusted and adjusted odds ratios (OR and aOR, respectively) were reported along with their respective 95% CIs. Significance was defined at the 5% level.

Out of the total number of Qataris who participated in the survey, 100% (n = 2,497) were included in STEP 1 and STEP 2, and 59% (n = 1,470) were included in STEP 3. As such, analyzed data were weighted to account for the differential selection probability and to have representative results for the population.

All analyses were conducted in IBM-SPSS Statistics version 26.

## Population attributable fraction of modifiable associated factors

The population attributable fraction (PAF) of the significant and modifiable factors associated with preDM, DM, and HTN, is the proportion of each outcome that can be theoretically prevented if the concerned factor is not present (while all other factors remain at their current levels), was estimated using Levin's formula [26,27]. As Levin's formula requires relative risk to compute this fraction, Cox regression with a constant time of 1, was used to estimate the adjusted relative risks (aRR) along with their 95% CIs [28]. Best- and worst-case scenarios for

each PAF were also estimated. For the best-case scenario, the lowest bounds of the 95% CIs for the prevalence of the associated factor and the aRR were inputted into the Levin's formula; while for the worst-case scenario, the highest bounds were used. This evaluation was conducted to inform prioritization of public health action.

# Results

## Sociodemographic and clinical characteristics

Table 1 presents the sample characteristics. Of the 2,497 participants, the majority were 18–44 years old (78.4%), reported having at least completed preparatory or secondary school (82.0%), were currently married (64.0%), and did not have parental consanguinity (58.2%). In addition, 40.0% of participants were physically inactive, 16.3% were current smokers, 14.3% consumed unhealthy diet, 41.5% were obese, and 66.2% were with central obesity. Majority reported having a family history of DM (67.0%) and HTN (64.4%).

There was a significant association between age and each of physical inactivity (p-trend<0.001), current smoking (p-trend = 0.009), obesity (p-trend<0.001), and central obesity (p-trend<0.001; S1 Fig). Physical inactivity, obesity, and central obesity all increased steadily with age.

## Prediabetes mellitus

The measured fasting blood glucose level ranged between 37 mg/dL and 326 mg/dL; thus, measurements were within the reading limits of the used POC device. Of the participants, 11.9% (95% CI 9.6%-14.7%) were living with preDM and almost half of them were <45 years old (Table 1). PreDM tended to increase with age for both women (p-trend = 0.008) and men (p-trend = 0.013; S2A Fig).

In univariate analysis, persons living with preDM were more likely to be ≥50 years old (OR = 3.45, 95% CI 1.72–6.91; p-value = 0.031), married (OR = 1.60, 95% CI 1.02–2.50; p-value = 0.039), obese (OR = 1.96, 95% CI 1.14–3.36; p-value = 0.016), and consuming unhealthy diet (OR = 1.93, 95% CI 1.05–3.56; p-value = 0.035; Table 2). Adjusted analyses showed that age ≥50 years (aOR = 2.32, 95% CI 1.15–4.66) and consumption of unhealthy diet (aOR = 1.84, 95% CI 1.01–3.36) were significantly associated with living with preDM. Sensitivity analysis including also sex in the multivariable model confirmed the results.

## Diabetes mellitus

Of the participants, 10.4% (95% CI 8.4%-12.9%) were living with DM (Table 1) and DM increased with age for both women (p-trend = 0.001) and men (p-trend = 0.001; S2B Fig).

In univariate analysis, persons living with DM were more likely to be ≥50 years old (OR = 10.01, 95% CI 5.47–18.57; p-value<0.001), with lower educational attainment (OR = 0.39, 95% CI 0.24–0.64 for preparatory or secondary school and OR = 0.43, 95% CI 0.26–0.73 for university; both relative to primary school or less; p-value<0.001), married (OR = 2.03, 95% CI 1.24–3.30; p-value = 0.005), physically inactive (OR = 1.95, 95% CI 1.3–2.93; p-value = 0.001), with central obesity (OR = 3.85, 95% CI 2.01–7.36; p-value<0.001), and living with HTN (OR = 3.50, 95% CI 2.40–5.13; p-value<0.001; Table 3). Adjusted analyses showed that age ≥50 years (aOR = 4.67, 95% CI 1.62–13.48), physical inactivity (aOR = 1.66, 95% CI 1.12–2.46), central obesity (aOR = 2.08, 95% CI 1.02–4.26), and HTN (aOR = 2.18, 95% CI 1.40–3.38) were significantly associated with being living with DM. Sensitivity analysis including also sex in the multivariable model confirmed the results.

**Table 1. Characteristics of the study sample.**

| Characteristics | | | All sample | | | Men N = 1,054; 49.7% (95% CI 46.8–52.6%) | | | Women N = 1,443; 50.3% (95% CI 47.4–53.2%) | | |
|---|---|---|---|---|---|---|---|---|---|---|---|
| | | | N | % | 95% CI | N | % | 95% CI | N | % | 95% CI |
| **Sociodemographic** | Age | ≤24 | 447 | 26.1 | 23.3–29.1 | 188 | 25.2 | 21.2–29.8 | 259 | 27.0 | 23.8–30.6 |
| | | 25–34 | 573 | 26.8 | 23.8–30.1 | 228 | 27.1 | 22.3–32.5 | 345 | 26.6 | 23.9–29.4 |
| | | 35–44 | 711 | 25.5 | 22.8–28.4 | 307 | 27.4 | 22-8–32.6 | 404 | 23.6 | 21.2–26.3 |
| | | 45–54 | 483 | 13.0 | 11.3–14.9 | 198 | 10.7 | 8.7–13.1 | 285 | 15.2 | 13.2–17.5 |
| | | 55–64 | 283 | 8.5 | 7.2–10.1 | 133 | 9.6 | 7.5–12.1 | 150 | 7.5 | 6.1–9.2 |
| | Education | Primary school or less | 506 | 18.0 | 15.1–21.3 | 170 | 14.3 | 11.4–17.8 | 336 | 21.6 | 18.0–25.7 |
| | | Preparatory or secondary school | 1115 | 51.4 | 48.0–54.9 | 512 | 55.1 | 50.2–59.8 | 603 | 47.8 | 44.0–51.6 |
| | | University | 875 | 30.6 | 26.6–35.0 | 372 | 30.6 | 25.5–36.2 | 503 | 30.6 | 26.2–35.4 |
| | Marital status | Non-married$ | 754 | 36.0 | 32.9–39.2 | 253 | 32.1 | 27.8–36.7 | 501 | 39.8 | 36.1–43.7 |
| | | Currently married | 1743 | 64.0 | 60.8–67.1 | 801 | 67.9 | 63.3–72.2 | 942 | 60.2 | 56.3–63.9 |
| | Parental consanguinity | No | 1566 | 58.2 | 54.3–61.9 | 649 | 56.5 | 51.2–61.7 | 917 | 59.8 | 55.2–64.2 |
| | | Yes | 931 | 41.8 | 38.1–45.7 | 405 | 43.5 | 38.3–48.8 | 526 | 40.2 | 35.8–44.8 |
| **Behavioral and physical** | Physical inactivity€ | No | 1441 | 60.0 | 56.1–63.7 | 711 | 68.6 | 63.6–73.3 | 730 | 51.4 | 46.1–56.7 |
| | | Yes | 1051 | 40.0 | 36.3–43.9 | 340 | 31.4 | 26.7–36.4 | 711 | 48.6 | 43.3–53.9 |
| | Smoking status | Non-smokers | 2035 | 79.3 | 76.7–81.6 | 620 | 59.8 | 54.9–64.5 | 1415 | 98.5 | 97.5–99.1 |
| | | Past smokers | 102 | 4.5 | 3.3–6.1 | 93 | 8.8 | 6.4–11.9 | 9 | 0.3 | 0.1–0.6 |
| | | Current smokers | 360 | 16.3 | 13.8–19.2 | 341 | 30.4 | 26.7–37.1 | 19 | 1.2 | 0.6–2.7 |
| | Diet | Healthy* | 1884 | 85.7 | 80.7–89.6 | 849 | 87.7 | 81.0–92.2 | 1035 | 83.6 | 78.2–87.9 |
| | | Unhealthy# | 310 | 14.3 | 10.4–19.3 | 120 | 12.3 | 7.8–19.0 | 190 | 16.4 | 12.1–21.8 |
| | Obesity status¥ | No | 1365 | 58.5 | 55.9–61.1 | 640 | 60.1 | 55.8–64.3 | 725 | 56.9 | 53.9–60.0 |
| | | Yes | 1086 | 41.5 | 38.9–44.1 | 395 | 39.9 | 35.7–44.2 | 691 | 43.1 | 40.0–46.1 |
| | Central obesity£ | No | 664 | 33.8 | 30.7–37.1 | 335 | 36.0 | 31.4–41 | 329 | 31.5 | 27.4–35.8 |
| | | Yes | 1672 | 66.2 | 62.9–69.3 | 687 | 64.0 | 59–68.6 | 985 | 68.5 | 64.2–72.6 |
| **Physiological** | DM family history& | No | 827 | 33.0 | 30.8–35.2 | 350 | 35.0 | 31.7–38.4 | 477 | 31.0 | 28.0–34.2 |
| | | Yes | 1670 | 67.0 | 64.8–69.2 | 704 | 65.0 | 61.6–68.3 | 966 | 69.0 | 65.8–72.0 |
| | HTN family history$ | No | 893 | 35.6 | 33.6–37.6 | 377 | 35.6 | 32.7–38.7 | 516 | 35.5 | 32.2–38.9 |
| | | Yes | 1604 | 64.4 | 62.4–66.4 | 677 | 64.4 | 61.3–67.3 | 927 | 64.5 | 61.1–67.8 |
| | PreDM | No | 1295 | 88.1 | 85.3–90.4 | 478 | 88.4 | 83.8–91.8 | 817 | 87.9 | 84.4–90.6 |
| | | Yes | 192 | 11.9 | 9.6–14.7 | 77 | 11.6 | 8.2–16.2 | 115 | 12.1 | 9.4–15.6 |
| | DM | No | 1285 | 89.6 | 87.1–91.6 | 476 | 89.0 | 84.8–92.1 | 809 | 90.1 | 87.6–92.2 |
| | | Yes | 186 | 10.4 | 8.4–12.9 | 77 | 11.0 | 7.9–15.2 | 109 | 9.9 | 7.8–12.4 |
| | HTN | No | 1562 | 67.1 | 64.4–69.6 | 696 | 71.5 | 67.9–74.9 | 866 | 62.7 | 58.2–66.9 |
| | | Yes | 903 | 32.9 | 30.4–35.6 | 346 | 28.5 | 25.1–32.1 | 557 | 37.3 | 33.1–41.8 |

$ Non-married defined as single, divorced, or widowed.

€ Physical inactivity defined as <150 minutes of moderate activity and <75 minutes of vigorous activity per week.

* Healthy defined as ≥5 servings of fruits and vegetables.

# Unhealthy defined as <5 servings of fruits and vegetables.

¥ Obesity defined as body mass index ≥30.

£ Central obesity defined as a waist circumference ≥94 cm for men and ≥80 cm for women.

& DM family history defined as at least one of the parents, children, brothers, or sisters is living with DM.

$ HTN family history defined as at least one of the parents, brothers, or sisters is living with HTN.

Abbreviations—CI: Confidence interval; HTN: Hypertension; DM: Diabetes mellitus; PreDM: Prediabetes mellitus.

**Table 2. Factors associated with prediabetes mellitus.**

| | | Crude/univariate | | | Multivariable | | | | | |
| | | | | | Primary analysis | | | Sensitivity analysis | | |
| | | OR | 95% CI | p-value | aOR | 95% CI | p-value | aOR | 95% CI | p-value |
|---|---|---|---|---|---|---|---|---|---|---|
| **Sex** | **Men** | *Reference* | | 0.870 | - | | | *Reference* | | 0.690 |
| | **Women** | 1.04 | 0.64–1.70 | | | | | 0.92 | 0.55–1.53 | |
| **Age** | **19–29** | *Reference* | | 0.031 | *Reference* | | 0.054 | *Reference* | | 0.055 |
| | **30–49** | 1.78 | 0.99–3.22 | | 1.34 | 0.75–2.41 | | 1. 34 | 0.75–2.43 | |
| | **50–64** | 3.45 | 1.72–6.91 | | 2.32 | 1.15–4.66 | | 2.33 | 1.15–4.72 | |
| **Education** | **Primary school or less** | *Reference* | | 0.082 | *Reference* | | 0.900 | *Reference* | | 0.910 |
| | **Preparatory or secondary school** | 0.55 | 0.32–0.93 | | 1.04 | 0.61–1.79 | | 1.03 | 0.60–1.75 | |
| | **University** | 0.70 | 0.41–1.22 | | 1.16 | 0.62–2.11 | | 1.13 | 0.62–2.05 | |
| **Marital status** | **Non-married[$]** | *Reference* | | 0.039 | *Reference* | | 0.800 | *Reference* | | 0.860 |
| | **Currently married** | 1.60 | 1.02–2.50 | | 1.06 | 0.70–1.60 | | 1.04 | 0.68–1.60 | |
| **Parental consanguinity** | **No** | *Reference* | | 0.580 | - | | | - | | |
| | **Yes** | 1.13 | 0.73–1.74 | | | | | | | |
| **Physical inactivity[€]** | **No** | *Reference* | | 0.420 | - | | | - | | |
| | **Yes** | 0.83 | 0.52–1.31 | | | | | | | |
| **Smoking status** | **Non-smokers** | *Reference* | | 0.390 | - | | | - | | |
| | **Past smokers** | 0.36 | 0.08–1.58 | | | | | | | |
| | **Current smokers** | 1.08 | 0.62–1.86 | | | | | | | |
| **Diet** | **Healthy[*]** | *Reference* | | 0.035 | *Reference* | | 0.048 | *Reference* | | 0.043 |
| | **Unhealthy[#]** | 1.93 | 1.05–3.56 | | 1.84 | 1.01–3.36 | | 1.86 | 1.02–3.37 | |
| **Obesity status[¥]** | **No** | *Reference* | | 0.016 | *Reference* | | 0.099 | *Reference* | | 0.100 |
| | **Yes** | 1.96 | 1.14–3.36 | | 1.57 | 0.92–2.68 | | 1.57 | 0.92–2.69 | |
| **Central obesity[£]** | **No** | *Reference* | | 0.060[€] | - | | | - | | |
| | **Yes** | 1.89 | 0.97–3.68 | | | | | | | |
| **DM family history[&]** | **No** | *Reference* | | 0.880 | - | | | - | | |
| | **Yes** | 0.97 | 0.62–1.51 | | | | | | | |
| **HTN family history[§]** | **No** | *Reference* | | 0.720 | - | | | - | | |
| | **Yes** | 0.93 | 0.61–1.41 | | | | | | | |
| **HTN** | **No** | *Reference* | | 0.250 | - | | | - | | |
| | **Yes** | 1.32 | 0.82–2.11 | | | | | | | |

[$] Non-married defined as single, divorced, or widowed.

[€] Physical inactivity defined as <150 minutes of moderate activity and <75 minutes of vigorous activity per week.

[*] Healthy defined as ≥5 servings of fruits and vegetables.

[#] Unhealthy defined as <5 servings of fruits and vegetables.

[¥] Obesity defined as body mass index ≥30.

[£] Central obesity defined as a waist circumference ≥94 cm for men and ≥80 cm for women.

[&] DM family history defined as at least one of the parents, children, brothers, or sisters is living with DM.

[§] HTN family history defined as at least one of the parents, brothers, or sisters is living with HTN.

[€] Central obesity was not included in the multivariable analysis because of collinearity with obesity, and because obesity had the higher crude OR.

Abbreviations—OR: Odds ratio; aOR: Adjusted odds ratio; CI: Confidence interval; HTN: Hypertension; DM: Diabetes mellitus.

## Hypertension

The measured systolic blood pressure ranged between 83 mmHg and 270 mmHg, while the diastolic blood pressure ranged between 50 mmHg and 170 mmHg. Of the participants, 32.9%

**Table 3. Factors associated with diabetes mellitus.**

| | | Crude/univariate | | | Multivariable | | | | | |
| | | | | | Primary analysis | | | Sensitivity analysis | | |
| | | OR | 95% CI | p-value | aOR | 95% CI | p-value | aOR | 95% CI | p-value |
|---|---|---|---|---|---|---|---|---|---|---|
| **Sex** | **Men** | *Reference* | | 0.558 | | - | | *Reference* | | 0.900 |
| | **Women** | 0.88 | 0.58–1.34 | | | | | 0.62 | 0.35–1.08 | |
| **Age** | **18–29** | *Reference* | | <0.001 | *Reference* | | <0.001 | *Reference* | | <0.001 |
| | **30–49** | 2.35 | 1.21–4.56 | | 1.71 | 0.61–4.77 | | 1.77 | 0.64–4.86 | |
| | **50–64** | 10.01 | 5.47–18.57 | | 4.67 | 1.62–13.48 | | 4.69 | 1.64–13.4 | |
| **Education** | **Primary school or less** | *Reference* | | <0.001 | *Reference* | | 0.250 | *Reference* | | 0.098 |
| | **Preparatory or secondary school** | 0.39 | 0.24–0.64 | | 0.82 | 0.45–1.48 | | 0.73 | 0.39–1.35 | |
| | **University** | 0.43 | 0.26–0.73 | | 0.63 | 0.36–1.09 | | 0.56 | 0.33–0.95 | |
| **Marital status** | **Non-married$** | *Reference* | | 0.005 | *Reference* | | 0.780 | Reference | | 0.970 |
| | **Currently married** | 2.03 | 1.24–3.3 | | 1.10 | 0.55–2.21 | | 1.02 | 0.50–2.07 | |
| **Parental consanguinity** | **No** | *Reference* | | 0.773 | | - | | | - | |
| | **Yes** | 0.94 | 0.59–1.48 | | | | | | | |
| **Physical inactivity€** | **No** | *Reference* | | 0.001 | *Reference* | | 0.012 | *Reference* | | 0.008 |
| | **Yes** | 1.95 | 1.3–2.93 | | 1.66 | 1.12–2.46 | | 1.73 | 1.16–2.58 | |
| **Smoking status** | **Non-smokers** | *Reference* | | 0.104 | *Reference* | | 0.520 | *Reference* | | 0.240 |
| | **Past smokers** | 1.68 | 0.62–4.53 | | 1.02 | 0.33–3.13 | | 0.78 | 0.23–2.68 | |
| | **Current smokers** | 0.59 | 0.33–1.03 | | 0.71 | 0.40–1.28 | | 0.56 | 0.29–1.09 | |
| **Diet** | **Healthy*** | *Reference* | | 0.590 | | - | | | - | |
| | **Unhealthy#** | 0.84 | 0.46–1.57 | | | | | | | |
| **Obesity status¥** | **No** | *Reference* | | 0.117¢ | | - | | | - | |
| | **Yes** | 1.37 | 0.92–2.05 | | | | | | | |
| **Central obesity£** | **No** | *Reference* | | <0.001 | *Reference* | | 0.044 | *Reference* | | 0.041 |
| | **Yes** | 3.85 | 2.01–7.36 | | 2.08 | 1.02–4.26 | | 2.09 | 1.03–4.25 | |
| **DM family history&** | **No** | *Reference* | | 0.376 | | - | | | - | |
| | **Yes** | 1.23 | 0.78–1.95 | | | | | | | |
| **HTN family history§** | **No** | *Reference* | | 0.316 | | - | | | - | |
| | **Yes** | 1.30 | 0.78–2.17 | | | | | | | |
| **HTN** | **No** | *Reference* | | <0.001 | *Reference* | | 0.001 | *Reference* | | 0.001 |
| | **Yes** | 3.50 | 2.4–5.13 | | 2.18 | 1.40–3.38 | | 2.20 | 1.38–3.51 | |

$ Non-married defined as single, divorced, or widowed.

€ Physical inactivity defined as <150 minutes of moderate activity and <75 minutes of vigorous activity per week.

* Healthy defined as ≥5 servings of fruits and vegetables.

# Unhealthy defined as <5 servings of fruits and vegetables.

¥ Obesity defined as body mass index ≥30.

£ Central obesity defined as a waist circumference ≥94 cm for men and ≥80 cm for women.

& DM family history defined as at least one of the parents, children, brothers, or sisters is living with DM.

§ HTN family history defined as at least one of the parents, brothers, or sisters is living with HTN.

¢ Obesity was not included in the multivariable analysis because of collinearity with central obesity, and because central obesity had the higher crude OR.

Abbreviations—OR: Odds ratio; aOR: Adjusted odds ratio; CI: Confidence interval; HTN: Hypertension; DM: Diabetes mellitus.

(95% CI 30.4%-35.6%) were living with HTN (Table 1) and HTN increased with age for both women (p-trend = 0.001) and men (p-trend = 0.001; S2C Fig).

In univariate analysis, persons living with HTN were more likely to be women (OR = 1.50, 95% CI 1.14–1.96; p-value = 0.004), ≥50 years old (OR = 8.40, 95% CI 5.64–12.50; p-

value<0.001), with lower educational attainment (OR = 0.40, 95% CI 0.31–0.52 for prepara-
tory or secondary school and OR = 0.50, 95% CI 0.37–0.67 for university; both relative to pri-
mary school or less; p-value<0.001), married (OR = 1.56, 95% CI 1.23–1.98; p-value<0.001),
smokers (OR = 1.75, 95% CI 0.99–3.08 for past smokers and OR = 0.74, 95% CI 0.50–1.10 for
current smokers; both relative to non-smokers; p-value = 0.023), obese (OR = 1.71, 95% CI
1.30–2.27; p-value<0.001), with central obesity (OR = 2.33, 95% CI 1.73–3.14; p-value<0.001),
and living with DM (OR = 3.33, 95% CI 2.29–4.84; p-value<0.001; Table 4). Adjusted analysis
showed that age ≥50 years (aOR = 6.35, 95% CI 3.06–13.18) and DM (aOR = 2.07, 95% CI
1.34–3.22) were significantly associated with living with HTN.

## Population attributable fraction of modifiable associated factors

Table 5 shows the PAFs, along with the best- and worst-case scenarios for each PAF, of the sig-
nificant and modifiable factors associated with preDM, DM, and HTN indicating that up to
7.7% of preDM cases can theoretically be prevented by reducing unhealthy diet; up to 14.9%,
39.8%, and 17.5% of DM cases can be prevented by reducing each of physical inactivity, central
obesity, and HTN, respectively; and up to 3.0% of HTN cases can be prevented by reducing
DM in the population.

## Discussion

The analyses presented herein characterized the epidemiological profiles of preDM, DM, and
HTN among Qatari adults by quantitatively assessing the prevalence of these outcomes and
associated risk factors. One in five persons had preDM or DM while one in three had HTN.
Consistent with Qatar and global evidence [1,7–14,29–34], the factors associated with such
conditions were age, central obesity, unhealthy diet, and physical inactivity. Earlier estimates
for the prevalence of each of preDM and DM [17,34] were further corrected by using the
appropriate preDM and DM cut-offs [18]. While the factors previously identified to be associ-
ated with DM in Qatar were affirmed [34], it was found that preDM was substantially underes-
timated (earlier estimate was 5.6% [17] compared to 11.9% herein) and DM was substantially
overestimated (earlier estimate was 16.7% [17] compared to 10.4% herein; Table 1). These
findings inform the national public health response towards an accurate allocation of resources
in the management and prevention of these disease outcomes.

This re-analysis of preDM and DM levels using the appropriate plasma-calibrated preDM
and DM cut-offs for the employed POC glucose measuring device confirms the earlier suspi-
cion by Lin et al. of incorrectly estimated preDM and DM prevalence levels [18]. Any future
policy, programming, health service planning, funding, and research deliberations and deci-
sions should factor these revised estimates rather than the earlier uncorrected estimates [17].

Evidence indicates that 5%-10% of preDM cases progress to DM annually if no adequate
and appropriate preventive interventions are implemented [35–38]. Consequently, with the
high burden of preDM in Qatar, DM prevalence is expected to increase considerably in the
coming years. Randomized clinical trials demonstrated that specific lifestyle and pharmaco-
therapy interventions targeting people living with preDM can reduce or delay progression
towards DM [39–45]. For instance, treatment of preDM with metformin reduced the risk of
developing DM by 30% [45]. Therefore, early detection of preDM followed by broad imple-
mentation of such lifestyle and pharmacotherapy interventions should be prioritized to better
manage preDM and thereby avert progression to DM.

With a preDM prevalence of 11.9% and DM prevalence of 10.4%, nearly a quarter of the
Qatari population is affected by hyperglycemia, one of the highest rates worldwide [1]. This
finding highlights the urgency of resource allocation and intensification of prevention

**Table 4. Factors associated with hypertension.**

| | | Crude/univariate | | | Multivariable | | |
|---|---|---|---|---|---|---|---|
| | | OR | 95% CI | p-value | aOR | 95% CI | p-value |
| **Sex** | **Men** | *Reference* | | 0.004 | *Reference* | | 0.059 |
| | **Women** | 1.50 | 1.14–1.96 | | 1.41 | 0.99–2.00 | |
| **Age** | **18–29** | *Reference* | | <0.001 | *Reference* | | <0.001 |
| | **30–49** | 1.94 | 1.45–2.59 | | 2.05 | 1.17–3.57 | |
| | **50–64** | 8.40 | 5.64–12.50 | | 6.35 | 3.06–13.18 | |
| **Education** | **Primary school or less** | *Reference* | | <0.001 | *Reference* | | 0.452 |
| | **Preparatory or secondary school** | 0.40 | 0.31–0.52 | | 0.89 | 0.66–1.21 | |
| | **University** | 0.50 | 0.37–0.67 | | 0.82 | 0.56–1.20 | |
| **Marital status** | **Non-married[$]** | *Reference* | | <0.001 | *Reference* | | 0.889 |
| | **Currently married** | 1.56 | 1.23–1.98 | | 0.96 | 0.66–1.39 | |
| **Parental consanguinity** | **No** | *Reference* | | 0.076 | *Reference* | | 0.070 |
| | **Yes** | 0.84 | 0.70–1.02 | | 1.33 | 0.98–1.80 | |
| **Physical inactivity[€]** | **No** | *Reference* | | 0.097 | *Reference* | | 0.875 |
| | **Yes** | 1.17 | 0.97–1.42 | | 1.03 | 0.73–1.44 | |
| **Smoking status** | **Non-smokers** | *Reference* | | 0.023 | *Reference* | | 0.103 |
| | **Past smokers** | 1.75 | 0.99–3.08 | | 2.16 | 1.06–4.40 | |
| | **Current smokers** | 0.74 | 0.50–1.10 | | 0.95 | 0.58–1.56 | |
| **Diet** | **Healthy[*]** | *Reference* | | 0.340 | - | | |
| | **Unhealthy[#]** | 1.22 | 0.84–1.77 | | | | |
| **Obesity status[¥]** | **No** | *Reference* | | <0.001[€] | - | | |
| | **Yes** | 1.71 | 1.30–2.27 | | | | |
| **Central obesity[£]** | **No** | *Reference* | | <0.001 | *Reference* | | 0.145 |
| | **Yes** | 2.33 | 1.73–3.14 | | 1.36 | 0.89–2.07 | |
| **DM family history[&]** | **No** | *Reference* | | 0.462 | - | | |
| | **Yes** | 1.10 | 0.86–1.41 | | | | |
| **HTN family history[$]** | **No** | *Reference* | | 0.104 | *Reference* | | 0.153 |
| | **Yes** | 1.18 | 0.97–1.44 | | 1.24 | 0.93–1.66 | |
| **PreDM** | **No** | *Reference* | | 0.634 | - | | |
| | **Yes** | 1.12 | 0.71–1.76 | | | | |
| **DM** | **No** | *Reference* | | <0.001 | *Reference* | | 0.001 |
| | **Yes** | 3.33 | 2.29–4.84 | | 2.07 | 1.34–3.22 | |

[$] Non-married defined as single, divorced, or widowed.

[€] Physical inactivity defined as <150 minutes of moderate activity and <75 minutes of vigorous activity per week.

[*] Healthy defined as ≥5 servings of fruits and vegetables.

[#] Unhealthy defined as <5 servings of fruits and vegetables.

[¥] Obesity defined as body mass index ≥30.

[£] Central obesity defined as a waist circumference ≥94 cm for men and ≥80 cm for women.

[&] DM family history defined as at least one of the parents, children, brothers, or sisters is living with DM.

[$] HTN family history defined as at least one of the parents, brothers, or sisters is living with HTN.

[€] Obesity was not included in the multivariable analysis because of collinearity with central obesity, and because central obesity had the higher crude OR.

Abbreviations—OR: Odds ratio; aOR: Adjusted odds ratio; CI: Confidence interval; HTN: Hypertension; DM: Diabetes mellitus; PreDM: Prediabetes mellitus.

programs, especially those targeting obesity. Obesity is by far the main driver of preDM and DM in Qatar and explaining 40% of DM cases (Table 5) [46], as well as HTN, physical inactivity, and unhealthy diet (Table 5). A previous analysis also affirmed that obesity and HTN were

**Table 5. Population attributable faction of modifiable factors associated with prediabetes mellitus, diabetes mellitus, and hypertension in Qatar.**

| Outcome | Modifiable factor | Prevalence of modifiable factor (%; 95% CI) | aRR (95% CI) | PAF (%; best- and worst-case scenario$) |
|---|---|---|---|---|
| **preDM** | | | | |
| | Unhealthy diet€ | 14.3 (10.4–19.3) | 1.59 (1.08–2.33) | 7.7 (0.9–20.4) |
| **DM** | | | | |
| | Physical inactivity* | 40.0 (36.3–43.9) | 1.44 (1.06–1.94) | 14.9 (2.3–29.2) |
| | Central obesity# | 66.2 (62.9–69.3) | 2.00 (1.17–3.43) | 39.8 (9.4–62.7) |
| | HTN | 32.9 (30.4–35.6) | 1.64 (1.19–2.27) | 17.5 (5.5–31.1) |
| **HTN** | | | | |
| | DM | 10.4 (8.4–12.9) | 1.30 (1.04–1.63) | 3.0 (0.4–7.5) |

$ For best-case scenario, the lowest bounds of the 95% CIs for the prevalence of the modifiable factor and the aRR were inputted into the Levin's formula; while for the worst-case scenario, the highest bounds were used.

€ Unhealthy diet defined as <5 servings of fruits and vegetables.

* Physical inactivity defined as <150 minutes of moderate activity and <75 minutes of vigorous activity per week.

# Central obesity defined as a waist circumference ≥94cm for men and ≥80cm for women.

Abbreviations—CI: Confidence interval; aRR: Adjusted relative risk; PAF: Population attributable fraction; PreDM: Prediabetes mellitus; DM: Diabetes mellitus; HTN: Hypertension.

major factors associated with DM in Qatar [34], thereby highlighting the urgency for evidence-based strategies to reduce associated morbidity and premature death. The findings also support that the high DM prevalence is a consequence of complex interactions between demographic, social, economic, and environmental factors [34].

With one in three living with HTN, the prevalence of this condition is high yet comparable to that observed in other countries regionally and globally [6,29–31]. HTN prevalence steadily increased with age consistent also with regional and global evidence [6,29–31]. Remarkably, HTN affected women more than men (Table 4), a similar finding to that observed in other Arab countries [6], but contrary to the global evidence where HTN prevalence is comparable for both sexes [30,31]. This finding remains to be explained. Despite its high prevalence, HTN awareness, prevention, and control programs remain rather limited in Qatar, but are critically needed along with population-based strategies for HTN prevention.

Limitations may have affected this study. While several factors were identified to be associated with the outcomes, we can only conclude an association but no causation using such cross-sectional data. For instance, both DM and HTN were found to be associated with each other, but the directionality cannot be established nor whether this is just a reflection of the underlying metabolic syndrome [47]. Fasting capillary blood glucose testing was used as opposed to fasting venous plasma glucose testing, which may bias assessed prevalence of preDM and DM [48]. With the definition of DM encompassing individuals taking DM medications on the day of the fasting blood glucose test, the estimated prevalence of DM may be overestimated considering that DM medications, such as metformin, can be administered to people not living with DM with the aim of regulating insulin level for other conditions, such as polycystic ovary syndrome [49,50]. Similarly, metformin is also used for preDM.

Recall bias and social desirability bias may have affected some measures such as reporting on unhealthy diet or physical inactivity, both of which are self-reported. Evidence suggests that self-reported physical activity is largely inflated relative to measurement using objective biomarkers [51]. Finally, while the PAF measure informs public health actions, it may not, in itself, provide information on the real risk and benefits of different public health interventions [52,53]. The PAF measure assumes that, following elimination of the risk factor, those

previously exposed to this risk factor will immediately attain the same disease risk of the unexposed, which might not be the case [52,53]. The PAF measure also assumes independence of the considered risk factor from other risk factors that influence disease risk so that it is possible to change only the population distribution of the considered risk factor while keeping the remaining fixed, which may not be necessarily the case [52,53].

The 2012 Qatar STEPwise Survey was conducted nine years ago [17], and the risk factor profiles and prevalence levels may have changed since the conduct of this survey. However, this survey remains the only nationally representative survey for DM in Qatar and continues to be the basis of policy, programs, health service planning, health promotion activities, and funding decisions, and thus the need to provide corrected estimates. Moreover, there is a tentatively planned new STEPwise survey sometime in the next few years. Our analyses provide the correct baseline for comparison to assess changes in outcomes and risk factors between this survey and the future survey.

In conclusion, one in five Qataris is living with preDM or DM and one in three is living with HTN, conditions that were found to be primarily driven by modifiable lifestyle factors including obesity, unhealthy diet, and physical inactivity. Control and management of DM and HTN should be a national priority in order to reduce the morbidity and mortality associated with CVDs. Lifestyle interventions targeting people living with preDM should also be a national priority to halt future progression to DM. Population-based prevention strategies addressing these three conditions simultaneously are essential to reduce the growing disease burden of these conditions and their associated disease sequelae [54].

## Supporting information

**S1 Fig. Age-distribution of physical inactivity, current smoking, obesity, and central obesity among Qataris aged 18–64 years old.**
(DOCX)

**S2 Fig. Age-distribution of prediabetes mellitus, diabetes mellitus, and hypertension among Qataris aged 18–64 years old.**
(DOCX)

**S1 Appendix. Glucose package insert 2010–2012 for the CardioChek PA analyzer used in this study.**
(PDF)

**S2 Appendix. English version of the "National STEP wise survey of chronic non-communicable diseases & risk factors instrument".**
(PDF)

## Acknowledgments

The authors are also grateful for infrastructure support provided by the Biostatistics, Epidemiology, and Biomathematics Research Core at Weill Cornell Medicine-Qatar.

## Author Contributions

**Conceptualization:** Laith J. Abu-Raddad.

**Data curation:** Mohammed H. Al-Thani, Kholood A. Al-Mutawa, Salah A. Alyafei, Muhammad A. Ijaz, Shamseldin A. H. Khalifa, Suresh B. Kokku, Amit C. M. Mishra, Benjamin V. Poovelil, Mounir B. Soussi, Amine A. Toumi.

**Formal analysis:** Mohammed H. Al-Thani, Kholood A. Al-Mutawa, Salah A. Alyafei, Muhammad A. Ijaz, Shamseldin A. H. Khalifa, Suresh B. Kokku, Amit C. M. Mishra, Benjamin V. Poovelil, Mounir B. Soussi, Amine A. Toumi, Soha R. Dargham, Susanne F. Awad, Laith J. Abu-Raddad.

**Funding acquisition:** Mohammed H. Al-Thani, Laith J. Abu-Raddad.

**Investigation:** Soha R. Dargham, Susanne F. Awad, Laith J. Abu-Raddad.

**Methodology:** Soha R. Dargham, Susanne F. Awad, Laith J. Abu-Raddad.

**Project administration:** Mohammed H. Al-Thani, Kholood A. Al-Mutawa, Salah A. Alyafei, Muhammad A. Ijaz, Shamseldin A. H. Khalifa, Suresh B. Kokku, Amit C. M. Mishra, Benjamin V. Poovelil, Mounir B. Soussi, Amine A. Toumi.

**Resources:** Mohammed H. Al-Thani, Kholood A. Al-Mutawa, Salah A. Alyafei, Muhammad A. Ijaz, Shamseldin A. H. Khalifa, Suresh B. Kokku, Amit C. M. Mishra, Benjamin V. Poovelil, Mounir B. Soussi, Amine A. Toumi.

**Supervision:** Mohammed H. Al-Thani, Laith J. Abu-Raddad.

**Visualization:** Soha R. Dargham, Susanne F. Awad, Laith J. Abu-Raddad.

**Writing – original draft:** Soha R. Dargham, Susanne F. Awad.

**Writing – review & editing:** Mohammed H. Al-Thani, Kholood A. Al-Mutawa, Salah A. Alyafei, Muhammad A. Ijaz, Shamseldin A. H. Khalifa, Suresh B. Kokku, Amit C. M. Mishra, Benjamin V. Poovelil, Mounir B. Soussi, Amine A. Toumi, Laith J. Abu-Raddad.

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
