## [Decision Letter · Decision Letter 0]

30 Jul 2021

PONE-D-20-35602

Characterizing epidemiology of prediabetes, diabetes, and hypertension in Qataris: a cross-sectional study

PLOS ONE

Dear Dr. Abu-Raddad,

Thank you for submitting your manuscript to PLOS ONE. After careful consideration, we feel that it has merit but does not fully meet PLOS ONE’s publication criteria as it currently stands. Therefore, we invite you to submit a revised version of the manuscript that addresses the points raised during the review process.

The reviewers felt that substantial changes must first be made to the manuscript. They cited concerns with the methodological approach and the presentation/justification of the statistical analysis. They also felt that the results and discussed should be more clearly presented. Their comments can be viewed in full, below.

We look forward to receiving your revised manuscript.

Kind regards,

Natasha McDonald, PhD

Academic Editor

PLOS ONE

Journal Requirements:

2. Please include additional information regarding the data extraction tool used in the study and ensure that you have provided sufficient details that others could replicate the analyses. For instance, if you developed a data extraction tool as part of this study and it is not under a copyright more restrictive than CC-BY, please include a copy, in both the original language and English, as Supporting Information, or include a citation if it has been published previously.

3. In your discussions and conclusions please take care to avoid statements implying causality from correlational research. For example, avoid the use of terms such as "predictors/ predictions" or “effects." Instead consistently use terms such as "associated with" or "associations.

4. Please describe how you were able to determine which was diagnosed first - hypertension or DM. This seems unlikely given your study design.

5. Under data availability, please provide the contact details for the body that is responsible for the data. For additional information on data availability policies, please review: https://journals.plos.org/plosone/s/data-availability#loc-acceptable-data-access-restrictions.

6. Thank you for stating the following financial disclosure:

This publication was made possible by NPRP grant number 10-1208-160017 from the Qatar National Research Fund (a member of Qatar Foundation). The statements made herein are solely the responsibility of the authors.

7. Please amend the manuscript submission data (via Edit Submission) to include author Mounir B. Soussi.

Reviewers' comments:

Reviewer's Responses to Questions

**Comments to the Author**

1. Is the manuscript technically sound, and do the data support the conclusions?

Reviewer #1: Yes

Reviewer #2: Yes

2. Has the statistical analysis been performed appropriately and rigorously? 

Reviewer #1: Yes

Reviewer #2: Yes

3. Have the authors made all data underlying the findings in their manuscript fully available?

Reviewer #1: Yes

Reviewer #2: No

4. Is the manuscript presented in an intelligible fashion and written in standard English?

Reviewer #1: Yes

Reviewer #2: Yes

5. Review Comments to the Author

Reviewer #1: Thank you for the opportunity to review this paper. I have some comments which you may find useful to incorporate or consider.

Firstly, I think this paper is valuable in that it corrects a previous erroneous calculation of the prevalence of type 2 diabetes. However, I think it should be made explicitly clearer in the text that your re-analysis – using verification of the POC glucose meter and test strips used were indeed plasma calibrated and requires a different glucose definition – confirms the suspicions made by the previous paper by Lin, et al., and that any future health service planning and/or research should use your revised calculations and not the previous WHO publication.

I think it would be useful to also explicitly make clear in the text that the POC glucose measuring device and test strips were plasma calibrated. This saves the reader having to dig through the Appendix. It would also help to make clearer why you have decided to re-analyse the prevalence of the other (non-glucose) risk factors. There is some justification for the re-analysis of preDM and DM (although I think this needs to be more clearly stated as I mentioned above) but it is not clear to me why you are re-analyzing blood pressure and obesity data. Is there also a concern about the calculations of prevalences of these risk factors in the original WHO report?

Finally, I have concerns about the usefulness of the data being published now considering the original measurements were taken 9 years ago. Population risk factor profiles can change substantially in that period of time. Why is re-using this dataset at this point in time useful? Is it simply to correct a past mistake that WHO made? To warn people about not using the previous incorrect calculations? Have there been major policy and/or funding decisions based on incorrect data that you wanted to highlight? How can you be sure that 2012 data is useful today for health service planning and health protection/promotion activities?

Reviewer #2: The authors did a tremendous amount of work but Major revisions are needed to improve the quality of the paper. The review comments are as attached in the file. Where the editor require clarification, I will be available to respond

6. PLOS authors have the option to publish the peer review history of their article (what does this mean?). If published, this will include your full peer review and any attached files.

Reviewer #1: No

Reviewer #2: **Yes: **Oyet Caesar

---

## [Author Response · Author response to Decision Letter 0]

4 Sep 2021

Editor’s comments

Comment: We thank the editor and editorial team for assessing our work and for the constructive feedback on our manuscript that enriched the article and improved its readability. Please find below a point-by-point reply addressing each of the editorial comments.

Answer: We have ensured that our manuscript meet the PLOS ONE style requirements.

2. Please include additional information regarding the data extraction tool used in the study and ensure that you have provided sufficient details that others could replicate the analyses. For instance, if you developed a data extraction tool as part of this study and it is not under a copyright more restrictive than CC-BY, please include a copy, in both the original language and English, as Supporting Information, or include a citation if it has been published previously.

Answer: We have now added a statement regarding the data extraction tool, cited the publication containing the information, and included the English version of the information in an Appendix (page 6 paragraph 3 of Methods and S2 Appendix).

3. In your discussions and conclusions please take care to avoid statements implying causality from correlational research. For example, avoid the use of terms such as "predictors/ predictions" or “effects." Instead consistently use terms such as "associated with" or "associations.

Answer: We have now removed the term “predictors” and avoided causal language (several instances throughout the manuscript).

4. Please describe how you were able to determine which was diagnosed first - hypertension or DM. This seems unlikely given your study design.

Answer: Indeed, with our study design we cannot determine which condition was diagnosed first. We have ensured that we use the term “association” to avoid statements implying causality (several instances throughout the manuscript).

5. Under data availability, please provide the contact details for the body that is responsible for the data. For additional information on data availability policies, please review: https://journals.plos.org/plosone/s/data-availability#loc-acceptable-data-access-restrictions.

Answer: The dataset of this study is a property of the Qatar Ministry of Public Health that was provided to the researchers through a restricted-access agreement that prevents sharing the dataset with a third party or publicly. Future access to this dataset can be considered through a direct application for data access to Dr. Mohammed H. Al-Thani, Director of Public Health Department, Ministry of Public Health (https://www.moph.gov.qa/english/Pages/default.aspx). Aggregate data are available within the manuscript and its Supplementary information. 

The statement above has now been added under the data availability section in the online submission system.

6. Thank you for stating the following financial disclosure: This publication was made possible by NPRP grant number 10-1208-160017 from the Qatar National Research Fund (a member of Qatar Foundation). The statements made herein are solely the responsibility of the authors. Please state what role the funders took in the study. If the funders had no role, please state: "The funders had no role in study design, data collection and analysis, decision to publish, or preparation of the manuscript." If this statement is not correct you must amend it as needed. Please include this amended Role of Funder statement in your cover letter; we will change the online submission form on your behalf.

Answer: The role-of-funder statement given above is correct and has now been added to the cover letter.

7. Please amend the manuscript submission data (via Edit Submission) to include author Mounir B. Soussi.

Answer: We have now added Mounir B. Soussi to the author list in the online submission system.

Reviewers’ comments

Reviewer #1: 

Thank you for the opportunity to review this paper. I have some comments which you may find useful to incorporate or consider.

Comment: We thank the reviewer for assessing our work and for the constructive feedback on our manuscript that enriched the article and improved its readability. Please find below a point-by-point reply addressing the reviewer’s comments.

1. Firstly, I think this paper is valuable in that it corrects a previous erroneous calculation of the prevalence of type 2 diabetes. However, I think it should be made explicitly clearer in the text that your re-analysis – using verification of the POC glucose meter and test strips used were indeed plasma calibrated and requires a different glucose definition – confirms the suspicions made by the previous paper by Lin, et al., and that any future health service planning and/or research should use your revised calculations and not the previous WHO publication.

Answer: We thank the reviewer for the suggestion. We have now explicitly stated this in the revised Introduction as suggested (page 5 paragraph 1 of Introduction). We have also highlighted the implications in the revised Discussion (page 14 paragraph 2 of Discussion).

2. I think it would be useful to also explicitly make clear in the text that the POC glucose measuring device and test strips were plasma calibrated. This saves the reader having to dig through the Appendix. It would also help to make clearer why you have decided to re-analyse the prevalence of the other (non-glucose) risk factors. There is some justification for the re-analysis of preDM and DM (although I think this needs to be more clearly stated as I mentioned above) but it is not clear to me why you are re-analyzing blood pressure and obesity data. Is there also a concern about the calculations of prevalences of these risk factors in the original WHO report?

Answer: We have now explicitly stated this in the revised Introduction as suggested (page 5 paragraph 1 of Introduction). 

As for the second point raised by the reviewer, we included the analysis of hypertension because the original WHO report estimated only the prevalence of hypertension and it did not examine its associations. In this study we complemented the earlier report with regressions to assess associations with hypertension. To address the reviewer’s comment, we have now added a clarification to this end in the revised Introduction (page 5 paragraph 2 of Introduction).

3. Finally, I have concerns about the usefulness of the data being published now considering the original measurements were taken 9 years ago. Population risk factor profiles can change substantially in that period of time. Why is re-using this dataset at this point in time useful? Is it simply to correct a past mistake that WHO made? To warn people about not using the previous incorrect calculations? Have there been major policy and/or funding decisions based on incorrect data that you wanted to highlight? How can you be sure that 2012 data is useful today for health service planning and health protection/promotion activities?

Answer: Indeed, as the reviewer has highlighted, this survey was conducted nine years ago, and the population risk factor profiles may have changed in such a period. However, this survey remains the only nationally representative survey for diabetes in Qatar and continues to be the basis of policy, programmatic, health service planning, health promotion activities, and funding decisions, and thus the need to provide corrected estimates. Moreover, there is a tentatively planned new STEPwise survey sometime in the next few years. Our analyses provide the correct baseline for comparison to assess changes in outcomes and risk factors between this survey and the future survey.

To address the reviewer’s comment, we have now highlighted this in the revised Discussion (page 16 paragraph 3 of Discussion).

Reviewer #2: 

The study looked at characterizing epidemiology of prediabetes, diabetes mellitus and hypertension in Qataris. The authors generated data from QATAR STEPWISE survey 2012. Major findings included increased prevalence of hyperglycemia and hypertension among the study participants. The authors did a tremendous work however, major and minor revisions are needed to improve the quality of the paper.

Comment: We thank the reviewer for assessing our work and for the constructive feedback on our manuscript that enriched the article and improved its readability. Please find below a point-by-point reply addressing the reviewer’s comments.

Major revisions

1. Results for several predictors were not sufficiently presented. There are no results for fasting blood glucose, blood pressure, waist/hip circumference and BMI. Though results for logistic regression exist, univariate analyses of these variables is very important.

Answer: We apologize for the confusion about inclusion of these results. We have now reported the odds ratios of the significant factors from the univariate analyses in the revised Results (page 11 paragraphs 2 and 4, and page 12 paragraph 3 of Results). All univariate results are further included in Tables 2-4.

2. Blood glucose test was performed using POC with limited maximum cut off. The authors didn't write in their paper what was done to samples with blood glucose levels higher than what the device could read. Where they excluded from the statistical analysis? The statements under limitations on this was not sufficient.

Answer: The glucose test system will detect glucose levels from 20 mg/dL to 600 mg/dL and will display a number value for results in this range. In our sample, the glucose level ranged between 37 mg/dL and 326 mg/dL which is within the reading of the device. 

To address the reviewer’s comment, we have now added these limits in the revised Methods, and reported the range found in the sample in the revised Results (page 8 paragraph 1 of Methods and page 11 paragraph 1 and page 12 paragraph 2 of Results).

3. Discussion section was not thorough. The authors directed their findings to what should be done but did not excisively discuss it. For example lines 219-221 under the discussion section stated that the study under estimated prevalence of prediabetes and over estimated prevalence of diabetes mellitus and no clear explanation was provided for the difference.

Answer: We thank the reviewer for the suggestions. We have now expanded the Discussion section to provide more implications as requested by the reviewer (page 14 paragraph 2, page 16 paragraph 3 of Discussion).

Minor revisions

4. I don't think I get the difference between p-trend and p-value, the authors may need to clarify this or be consistent since there was no mention of such under methods section.

Answer: We apologize for the confusion. P-trend denotes the p-value for the linear trend analyses in the study. We have now added this clarification in the revised Methods (page 9 paragraph 1 of Methods).

5. Lines 177, 186 and 197 mentioned "univariable" and I think it should be univariate.

Answer: This has now been implemented in the revised manuscript (several instances throughout the manuscript). 

6. Lines 146 and 147 under results section has N=2497 or similar and since this represent samples it should be n=,....... 

Answer: We thank the reviewer for pointing this out. This has now been implemented in the revised Methods as suggested (page 9 paragraph 2 of Methods).

7. You stated prevalence of hyperglycemia rather than diabetes or prediabetes yet hyperglycemia is a state and not a disease and sometimes we have transferred hyperglycemia secondary to other non-diabetic causes. The authors may need to edit or revise that.

Answer: The term “hyperglycemia” has now been removed and replaced by “diabetes or prediabetes” as suggested (several instances throughout the manuscript).

---

## [Decision Letter · Decision Letter 1]

14 Oct 2021

Characterizing epidemiology of prediabetes, diabetes, and hypertension in Qataris: a cross-sectional study

PONE-D-20-35602R1

Dear Dr. Laith J. Abu-Raddad,

We’re pleased to inform you that your manuscript has been judged scientifically suitable for publication and will be formally accepted for publication once it meets all outstanding technical requirements.

Kind regards,

Fernando Guerrero-Romero, MD, PhD

Academic Editor

PLOS ONE

Additional Editor Comments (optional):

No additional comments

Reviewers' comments:

Reviewer's Responses to Questions

**Comments to the Author**

1. If the authors have adequately addressed your comments raised in a previous round of review and you feel that this manuscript is now acceptable for publication, you may indicate that here to bypass the “Comments to the Author” section, enter your conflict of interest statement in the “Confidential to Editor” section, and submit your "Accept" recommendation.

Reviewer #1: All comments have been addressed

Reviewer #2: All comments have been addressed

2. Is the manuscript technically sound, and do the data support the conclusions?

Reviewer #1: Yes

Reviewer #2: Yes

3. Has the statistical analysis been performed appropriately and rigorously? 

Reviewer #1: Yes

Reviewer #2: Yes

4. Have the authors made all data underlying the findings in their manuscript fully available?

Reviewer #1: Yes

Reviewer #2: Yes

5. Is the manuscript presented in an intelligible fashion and written in standard English?

Reviewer #1: Yes

Reviewer #2: Yes

6. Review Comments to the Author

Reviewer #1: I have no further comments for the authors as they have addressed my previous queries in the first review.

Reviewer #2: All questions were addressed approximately by the authors and the manuscript can now be accepted for publication.

7. PLOS authors have the option to publish the peer review history of their article (what does this mean?). If published, this will include your full peer review and any attached files.

Reviewer #1: No

Reviewer #2: **Yes: **Caesar Oyet

---

## [Editor Report · Acceptance letter]

18 Oct 2021

PONE-D-20-35602R1 

Characterizing epidemiology of prediabetes, diabetes, and hypertension in Qataris: a cross-sectional study 

Dear Dr. Abu-Raddad:

I'm pleased to inform you that your manuscript has been deemed suitable for publication in PLOS ONE. Congratulations! Your manuscript is now with our production department. 

Kind regards, 

on behalf of

Dr Fernando Guerrero-Romero 

Academic Editor

PLOS ONE